# Structural Damage Detection Using an Unmanned Aerial Vehicle-Based 3D Model and Deep Learning on a Reinforced Concrete Arch Bridge

**Mary C. Alfaro [1], Rodrigo S. Vidal [1], Rick M. Delgadillo [2,*], Luis Moya [2] and Joan R. Casas [3]**

[1] Department of Civil Engineering, Universidad Peruana de Ciencias Aplicadas (UPC), Prolongación Primavera 2390, Monterrico, Santiago de Surco, Lima 15023, Peru; u202010564@upc.edu.pe (M.C.A.); u202016374@upc.edu.pe (R.S.V.)

[2] GERDIS Research Group, Department of Engineering, Civil Engineering Division, Pontificia Universidad Católica del Perú—PUCP, Av. Universitaria 1801, Lima 15088, Peru; lmoya@pucp.edu.pe

[3] Department of Civil Engineering and Environmental Engineering, Technical University of Catalonia (Barcelona Tech), Campus Nord, C1 Building, Jordi Girona, 1-3, 08034 Barcelona, Spain; joan.ramon.casas@upc.edu

\* Correspondence: rick.delgadillo@pucp.pe

**Abstract:** Visual inspection is a common method for detecting structural damage, but has limitations in terms of subjectivity, time, and access. This research proposes an innovative approach to identify cracks using a 3D model generated from photographs of an unmanned aerial vehicle (UAV) and the use of a convolutional neural network (CNN). These networks are effective in detecting complex patterns, improving the accuracy and efficiency of damage identification based on simple visual inspection. The case study is the old Villena Rey bridge in Lima, Peru. The methodology covers (i) the development of a 3D model of the bridge structure, (ii) the extraction of photographs of the model and its binary segmentation, (iii) the application of deep learning through the training and testing phase of a CNN to achieve crack detection in photographs, and (iv) damage location within the 3D model. An 88.4% accuracy was achieved in crack detection, identifying 18 damage points, of which 3 turned out to be false positives. Additionally, it was determined that the left pillar in the southern area of the bridge presented the highest concentration of damage, which underlines the effectiveness of the method used.

**Keywords:** structural damage detection; bridge; 3D model; binary segmentation; deep learning; convolutional neural networks

## 1. Introduction

Bridges are important and relevant structures for the road infrastructure of every city. However, reasons such as lack of maintenance, environmental conditions, and external loads generates their deterioration and even their sudden collapse [1]. According to a statistical investigation, it was determined that the main causes of damage to bridges are generated by errors in design and construction, overloading, collisions, and natural phenomena [2]. Therefore, the detection of structural damage in bridges is a useful and necessary practice.

Currently, the detection method is mainly based on visual inspections carried out by a team of specialized professionals [3]. This process must be executed following the rules and guidelines of each government or agency [4]. These specialists must confirm the appearance of damage such as cracks, concrete chips, and excessive deformations, among other deteriorations that can be seen with the naked eye [5,6]. However, it is a

limited and subjective method due to the complexity of the damages that can be detected and the different structural characteristics that may exist in a bridge [3]. Certain damages may not be recognized by the human eye or, due to their location, may not be easily seen. Consequently, in recent years, various researchers have proposed technological methods to achieve more reliable results and more objective and agile processes. These are usually based on the construction of digital models and processing of visual data (images) deep learning networks that can help detect damage. On the one hand, the construction of a 3D model has become a frequent practice for this purpose [7]. In the absence of graphical information of the bridge, the way to build a 3D model of an existing asset is usually through photogrammetry with images obtained with UAVs, which allows us to obtain visual data that are difficult to access with the human eye or traditional cameras [8]. Within these images, the existence of damage can be determined. Therefore, the model will have information on their existence and location. However, one way to complement this damage detection method is through deep learning. This technique is based on Artificial Intelligence (AI) and results in the configuration of CNNs. These detect, identify, and/or segment some desired element within a group of data. Previously, the CNN must receive a database and instructions so that it is trained to recognize a desired pattern and can detect what is required. Subsequently, it receives input data other than the training data, performs "feature extraction", and returns the results with a certain degree of precision [3]. If there is a bank of photographs that can show the presence or absence of damage in the different parts of a bridge, a network can be trained to precisely detect in which images damage is evident.

Below is the literature on the use of UAVs to generate 3D models and methods, such as deep learning, for damage detection. There are authors who detect damage visually within a 3D model without incorporating such sophisticated computational methods. However, they determine that the use of UAVs represents an advantage over traditional inspections due to the wide access to all parts of the structure. Perez Jimeno et al. [8] carried out UAV flights to build a 3D model of the Río Claro Bridge (Colombia) and, in this way, detected damage that had not been visually detected in a previous inspection. Congress et al. [7] made 3D models of the Montana Creek, Canyon Creek, and Eagle River Southbound bridges (Alaska, United States) and detected rust, worn paint, and cracks, among others. However, although 3D models will have information about the presence of damage, a human must still continue to detect them visually within the model. For this reason, there are authors who combined model construction with computational methods. Potenza et al. [9] implemented a software called "Deep" to detect cracks in images through processing. The cracks with a different color are detected using rules programmed into the software. The authors later used deep learning in order to detect damage. In the literature, first of all, Pantoja-Rosero et al. [3] used a CNN to process images obtained with UAVs, detect any cracks that may present, and be able to capture the building and its damage in a 3D model. As a result of the investigation, an average of 68% of cracks present in the structure were identified. Secondly, Kong and Hucks [5] created two 3D models of historical buildings at different times using UAV and photogrammetry, which are in a constant state of deterioration. They used a CNN that can recognize the differences between both models, which could be damage. As a result, cracks, loose materials in the structure, and settlement in the ground were detected. In the case of concrete bridges, Kim et al. [10] focused their research on CNNs for processing photographs and detecting damage such as cracks, efflorescence, breakage, water leakage, material segregation, and exposure of steel bars, which was achieved in Bridge D (Gangwon, Korea). To improve crack quantification accuracy, the bridge had markers installed, and the UAV had to be precisely positioned on each part of the concrete element. Finally, Li et al. [11] suggests using high-resolution cameras and testing various

orientations at the UAV location to achieve better results. Other authors delve into the speed of obtaining data through photogrammetry and its communication and transfer with a developed 3D model. There is research based on "AIoT", a concept known as "the Internet Of Things", which refers to the connectivity between objects and the rapid collection of information. Gao et al. [12] use UAVs to collect data and "AIoT" to transfer it to a 3D model as part of their proposal for periodic inspections of reinforced concrete bridges. Also, there are researchers who "combine" the 3D model obtained from the field with other types of modeling. Generally, the damage detected is "visual", and its location in the digital twin will allow for this damage to be related to some internal component of the structure, which can be observed in the second incorporated model. Wang et al. [4] constructed a digital twin composed of a finite element model and a 3D model using UAV and photogrammetry. The developed CNN detects the areas with the highest density within the "point cloud" of the 3D model, which indicates a crack, which is located within the finite element model. Levine and Spencer [6] used a digital twin composed of a BIM model and a 3D model, trained a CNN to identify damage in the images obtained with the UAV, and, by locating its damage within the 3D model, it also locates damage within the BIM model. Their case study was a reinforced concrete building, and they detected moderate and severe damage to non-structural walls and displacements in beam–column connections.

After an exhaustive review of the literature, methods based on the construction of 3D models and the use of deep learning have been effective and versatile for detecting structural damage in infrastructure, such as bridges and buildings. These advances suggest significant potential to improve current practices in structural integrity assessment. However, a common aspect in research is that, for the application of deep learning, the UAV is required to have high-precision cameras and to be located in a way that best captures damage [11]. In the field, this condition can be limiting, since the flight of the drone is required to be as fast, safe, and agile as possible. However, in this way, the captured images may have unforeseen "noise", as unwanted elements could be captured in the background of the image. In turn, it is possible that the possible visual damage does not stand out due to the quality of the satellite signal, the required distance between the drone, and the structure and the environmental conditions of the area. A solution to this is data pre-processing, which reduces unnecessary information in the images and highlights the crack if there is one within the photograph, which is achieved with binary segmentation [13]. With these data already "delimited", a neural network can begin its processing in a more fluid way.

Therefore, this research will focus on the Villena Rey Bridge, located in the Miraflores district in Lima, Peru, as a case study. A method is proposed for the detection of structural damage, specifically cracks, by generating a three-dimensional model and using deep learning. The digital model of the bridge will be obtained using a UAV to capture detailed images. Subsequently, a process of eliminating "noise" present will be carried out using binary segmentation algorithms. Then, a convolutional neural network (CNN) will be developed that will allow for cracks to be detected. To do this, said network will be trained with a large database and indications to then perform damage detection with the images previously obtained using the UAV. Finally, the damage will be located within the 3D model.

*Case Study*

To test the proposed method, the Bridge Villena Rey (Figure 1), which is located in the Miraflores district in Lima, Peru, is taken as a case study. This bridge dates back to 1967 and passes over the "Bajada Balta" ravine. It has a two-lane road for vehicular passage and, years ago, it was bidirectional. However, due to its high traffic volume, the bridge has had traffic congestion problems and, therefore, a twin bridge was built in 2014—that is,

a bridge with similar characteristics was built next to it, a new two-lane road bridge [14]. With this modification, the lanes of the Villena Rey Bridge (old) began to be used for one direction, and the lanes of the "twin" bridge began to be used for the other. Its location has high pedestrian and vehicular traffic, and it is in a medium-density residential area, also, with commercial and public recreation [15]. Therefore, due to its intensive use and high vehicular traffic, it is exposed to structural damage. In addition, due to its geographical location, it has the risk of suffering damage from natural causes typical of Lima, such as seismic events and humidity, among others.

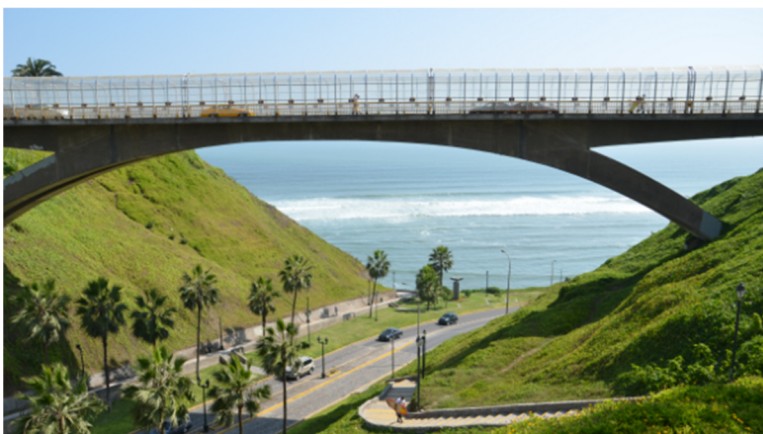

**Figure 1.** Villena Rey Bridge.

As part of this investigation, a visual inspection was performed on the old bridge, as would be performed in a traditional structural damage detection. In this way, it was confirmed that the bridge is damaged. The results of the inspection are detailed in Figure 2 and Table 1. When the width of the crack is greater than 0.4 mm, there is already a high level of impact, which indicates a reduction in the earthquake–resistant capacity [16]. The method proposed in this research is able to recognize damages that are at least 0.4 mm wide or even larger, due to the distance of the UAV from the structural elements, which is usually from 2 to 3 m. The UAV would not be able to capture damages narrower than 0.4 mm. Therefore, only cracks with structural significance are captured.

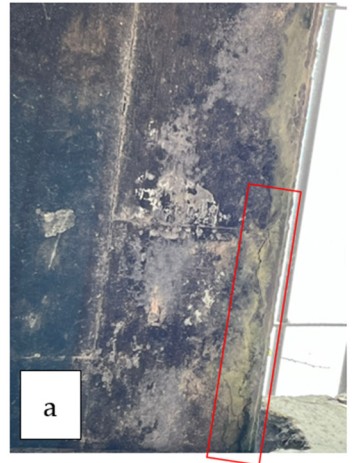 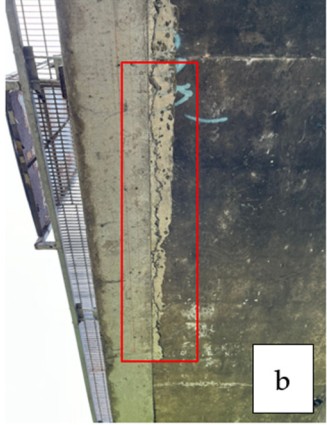

**Figure 2.** *Cont.*

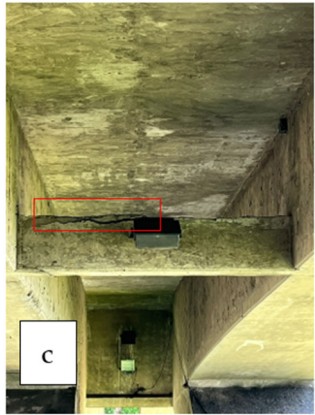
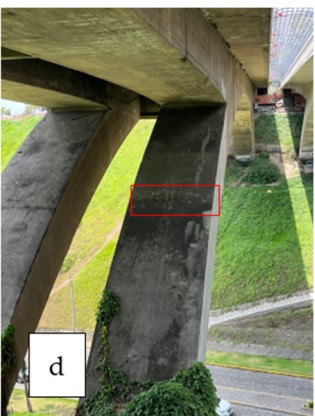

**Figure 2.** Structural cracks and fissures detected in a previous visual inspection. (**a**) Crack identified under bridge deck. (**b**) Crack identified under bridge deck. (**c**) Crack identified in cross-beam. (**d**) Crack identified in arch.

**Table 1.** Fissures and cracks reported visually in a technical inspection on the Villena Rey Bridge, visually estimated width, and its level of impact on the structure.

| Damage | Ubication | Specification | Visually Estimated Slit Width (mm) | Width Classification | Level of Impact on the Structure |
|---|---|---|---|---|---|
| 1 | Slab | Cleft, crack | 1 mm | Crack | Very high |
| 2 | Slab | Cleft, crack | 2 mm | Crack | Very high |
| 3 | Beam | Cleft, crack | 3 mm | Crack | Very high |
| 4 | Pier | Cleft, crack | 3 mm | Crack | Very high |

## 2. Materials and Methods

### 2.1. Framework for Damage Detection

This section details the framework developed for damage detection improvements in the Villena Rey Bridge, which is based on the use of digital models and deep learning techniques for the development of convolutional neural networks (CNNs) capable of automatically detecting obvious cracks. The sequence of processes proposed in this study is illustrated in Figure 3. First, a three-dimensional digital model is built using the Agisoft Metashape software (1.8.2. version) based on a photogrammetric survey carried out with a UAV. Secondly, binary segmentation processing is performed. For this step, it must be taken into account that the 3D model is made up of hundreds of photographs of the Villena Rey Bridge that may contain unwanted noise or may not focus very well on possible cracks, due to the agility at which the UAV flew over the area, capturing images. Finally, binary segmentation must be performed—that is, converting it to pure black and white with the intention that the noise is eliminated or reduced and that the crack is seen as a fully visible and highlighted line. To do this, these photographs are converted to grayscale images and, with the threshold method using the OTSU technique, a reference "threshold" value is obtained to determine which grayscale turns white and which turns black.

Thirdly, it should be considered that the subsequent step is the application of the CNN that considers two large phases: training, and the final test phase. On the one hand, for training, the CNN must be fed two large groups of images and binaries: one containing a crack, and one not. In this way, it can extract the features of both groups for the subsequent process. For the first training group, the binary segmentation process has been carried out on images from other databases containing cracks and another database of binary images that have already been taken. For the second group, only some binary images

generated from the photographs extracted from the 3D model of the bridge that do not contain cracks have been taken. On the other hand, for the final phase test, where the CNN must already be able to detect damage to the object of study, the input data for the CNN were the processed binary images from the photographs of the Villena Rey Bridge that were not used in the training phase, without distinction as to whether they contain cracks or not. Finally, the CNN detects which binary images have cracks and which do not, obtaining a percentage of accuracy. Finally, since the images have labels, a record will be generated, indicating which of them have cracks according to the CNN analysis, and, using these labels, these damages can be located within the 3D model.

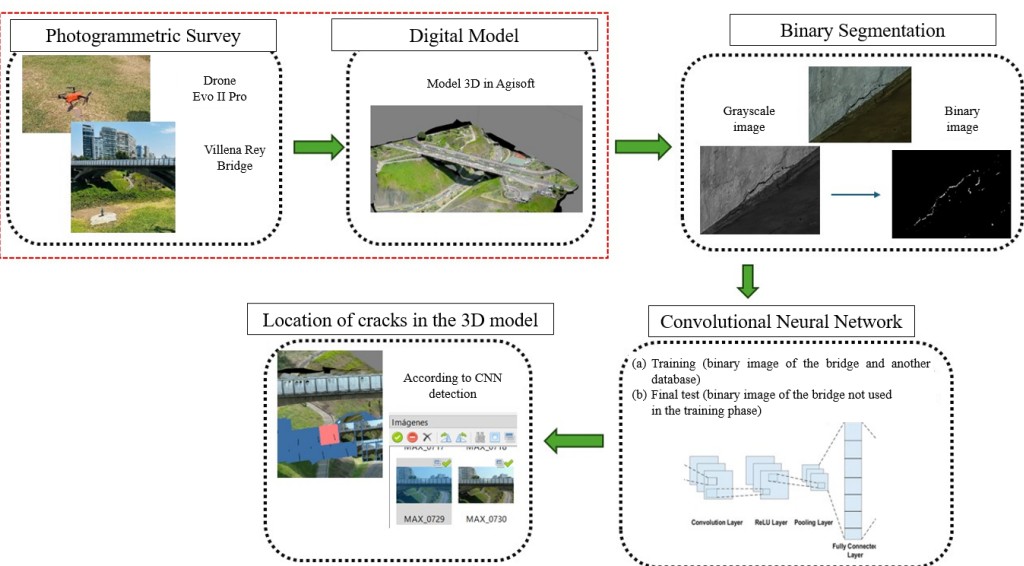

**Figure 3.** Overall research framework.

As a result of the investigation, a report of the damage located in the model will be obtained, as well as the precision and degree of reliability of the CNN in detecting the damage. When taking a photograph, the UAV has information about its exact location, which is necessary to carry out a "photogrammetry" process. This location is captured in the 3D model in the Agisoft software. In this way, it is possible to see which element of the bridge a photograph refers to.

Below, the main processes that make up the proposed methodology are described, as well as the important considerations for its implementation.

### 2.2. Digital Model

A digital model is a virtual representation of a physical and real element [17]. In the world of engineering, various physical assets have been recreated on computers—That is, 3D models have been created for various purposes: making predictions based on simulations, analyses, and visualizations of nature [18].

For this investigation, a 3D model of the Villena Rey Bridge is built, which is generated from images captured through a photogrammetric survey using a UAV. Photogrammetry is a method of generating visual information through images that provide information on the sizes, shapes, and positions of the elements it captures. These images also provide in-depth information, which generates a cloud of points so that, by processing and making them compatible, a detailed 3D model can be created.

This model visually represents the bridge and provides information on its geometry, appearance, and details of its structural elements. Furthermore, as it is made up of images of the bridge, it will provide an extensive and organized database of photographs, which

will be located within the physical space of the model—that is, each photograph captures a specific area of a specific element of the bridge, and said location can be seen within the digital representation.

### 2.2.1. Flight of the Unmanned Aerial Vehicle and Photogrammetric Survey of the Villena Rey Bridge

For the photogrammetric survey, an "EVO II Pro" drone from the Drone Professional Perú brand is used (Figure 4). Among its specifications, it stands out that it must have an adequate mass so that it is not destabilized by the force of the air (1174 g), adequate ascent and descent speeds (5 m/s and 3 m/s, respectively, in its standard mode), a long flight distance (25 km), and a safe maximum flight time (40 min). The camera should take good quality photos and produce images of various sizes (5472 × 3648; 5472 × 3076; 3840 × 2160). The data mentioned refer to the model and brand of the drone to be used. The approximate distances for taking photographs of the structural elements of the bridge are between 2 and 3 m. It is advisable not to bring the UAV too close to the bridge, as it may destabilize and collide due to air turbulence.

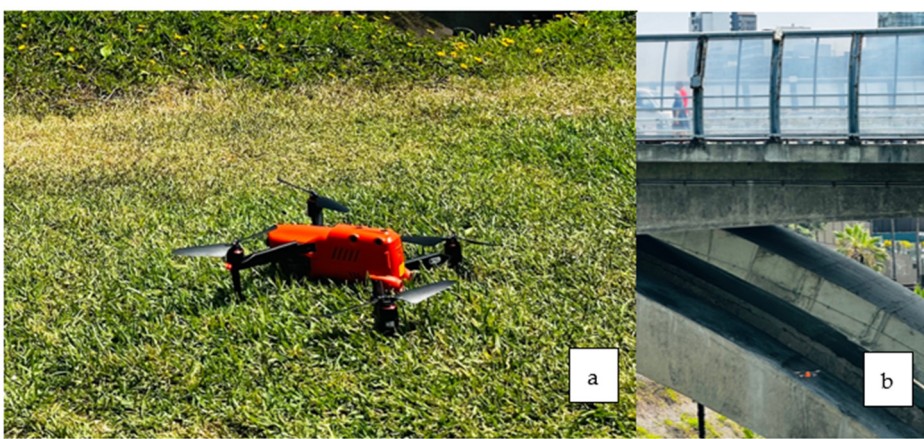

**Figure 4.** (**a**) Evo II Pro, Drone Professional Peru; (**b**) photogrammetry using UAV.

Prior to the flight of the equipment, the Region of Interest (ROI) must be delimited—that is, delimit the space and the elements to photograph. In this case, the objective was to photograph the entire bridge and emphasize the structural elements. In addition, the flight must be carried out on a sunny day, with little presence of violent winds and with filters on the camera so that it can capture images with good resolution.

### 2.2.2. Construction of the Digital Model in Agisoft Software

Once the flight has been completed, the camera will have captured a large number of photographs, which will be uploaded to the Agisoft Metashape software to carry out the photogrammetric processing with the following procedure:

(a) Generate a sparse point cloud with the "photo alignment" function.
(b) Build a general dense point cloud, considering the information from each image, integrating them into a single model.
(c) Build a 3D mesh using the previous dense point cloud as a source of data.
(d) Build a texture on the 3D mesh to create the digital model.

It is worth mentioning that the trajectory that the UAV followed was close to the bridge from its lateral sides and from its upper side. The UAV has taken 5 trajectories, which capture various parts of the bridge, whose specifications are shown in Table 2.

It is possible that there are integration problems with some groups of images due to the location of the camera in which they were captured. Sometimes, the correct idea is to obtain a

group of images that are not within the main group, since they could be required to create a separate "sub model". In these cases, a model can be composed of independent blocks.

For the case study, a 3D model is obtained that is composed of 5 blocks, with 1 visually representing certain elements of the bridge. The summary of what was obtained in the five blocks is shown in Figure 5 and Table 2. Blocks 2, 3, and 4 are those that contain suitable images for subsequent processing of damage detection, since they capture the presence or absence of cracks in the concrete elements of the bridge. Blocks 1 and 5 have very general images of the environment, which are used for the construction of the 3D model of the bridge and a general view of it. Therefore, blocks 2, 3, and 4 will be the visual data providers for the following processes.

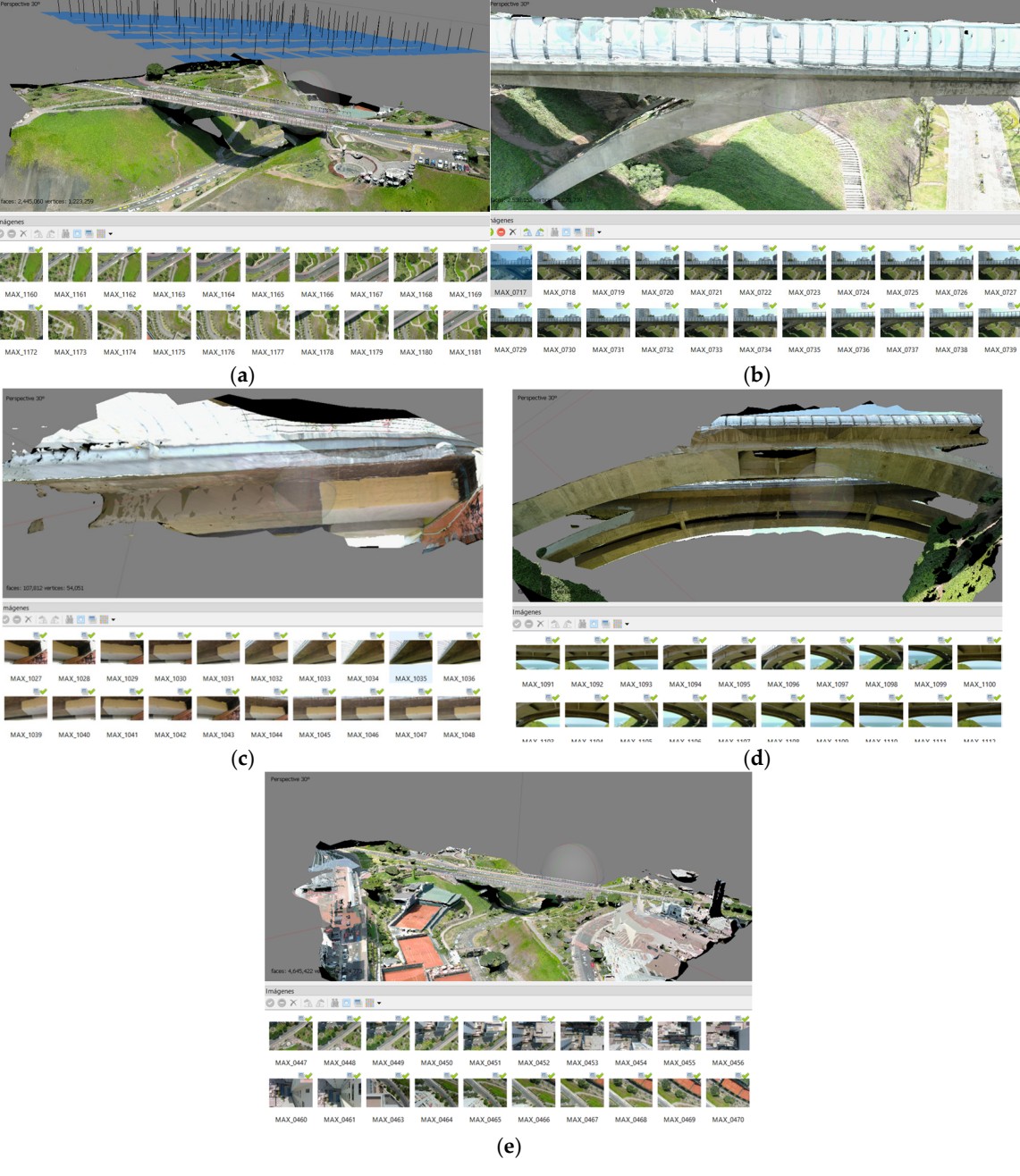

**Figure 5.** Summary of the blocks that constitute the digital model of the Villena Rey Bridge (display from software). (**a**) Block 1: Perpendicular photos. (**b**) Block 2: Side photos 1. (**c**) Block 3: Side photos 2. (**d**) Block 4: Side photos 3. (**e**) Block 5: Wide lift.

**Table 2.** Summary of the blocks that constitute the digital model of the Villena Rey Bridge.

| Block | Number of Photographs | Number of Points | Area They Cover |
|---|---|---|---|
| Block 1 | 105 | 95610 | Provides visual and geometric information on the location and shape of the bridge with the environment |
| Block 2 | 268 | 94246 | Provides visual information on the left side of the board, the beam main left, and the arch-shaped pillar of the bridge |
| Block 3 | 64 | 53427 | Provides visual information of the right side of the main beam right |
| Block 4 | 65 | 29922 | Gives visual information of the bottom of the main beams and arch-shaped pillars |
| Block 5 | 268 | 197326 | Provides visual information about the wide environment in which the device is located on the bridge |

The photographs are labeled and located within the model. When photographs with damage are detected, they can be located within the model, which will provide information on the location of the damage on the bridge.

### 2.2.3. Binary Image Segmentation

As reviewed in the literature, it is observed that in most crack detection investigations using image processing, they tend to restrict the field of view of the photographs to the damage. However, in this research, an agile flight by the UAV is proposed, which results in images captured from considerable distances that contain not only the cracks, but also the additional elements of the bridge or its surroundings. If these photographs were processed in a CNN, the network would not be able to detect the damage easily. Therefore, an image must be obtained that omits unwanted information and highlights the damage, which is achieved by binary segmentation in the Matlab software (2023 version).

The segmentation process in RGB images has become one of the most used techniques in the field image analysis. RGB color spaces are made up of three-color bands, each with intensity values from 0 to 255. Full intensity (255, 255, 255) results in white, while no intensity (0, 0, 0) results in black [19]. The main reason is that it provides the basis for pattern recognition and its understanding since it divides images into regions with different characteristics and extracts regions of interest [20]. There are different techniques that comprise the development of binary segmentation: thresholding, edge detection, clustering, methods based on partial equations, graph partitioning methods, semi-automatic segmentation, and multiscale segmentation, among others [13].

After extracting images from the 3D model, and in the context of the proposed solution, it is suggested to use the thresholding technique to generate binary images. This is because it works with a parameter called a threshold that determines which grayscale of an image turns to black or white and can be adjusted to suit the research. In their research, [21] managed to eliminate 99% of impurities from images that capture cracks in reinforced concrete, applying this method. Previously, it is necessary to convert the images to grayscale. To do this, algorithms will be implemented that allow for both processes to be carried out in a unified manner—namely, the conversion to grayscale, and the subsequent binarization of the image.

### 2.2.4. Grayscale

Before applying the threshold method, the RGB image must be converted to grayscale. This process is important since an RGB image, having a three-dimensional property, makes its processing heavy. Therefore, converting the image to grayscale means converting all color information to grayscale [22].

An image is made up of 24-bit wide pixels distributed as follows: 8 bits for the color red, 8 bits for the color blue, and 8 bits for the color green. Therefore, in the grayscale transformation, the pixel only contains 8 bits that provide information about brightness, but not color. Mathematically, there are different methods that are used to obtain a grayscale image, and among these, we have the following: average method, luminosity method, desaturation method, and decomposition method. Among the different existing methods, the one used for algorithm functions, such as "rgb2gray" in Matlab, is the luminosity method [23]. As an example, two results of converting images to grayscale are evident in Figure 6.

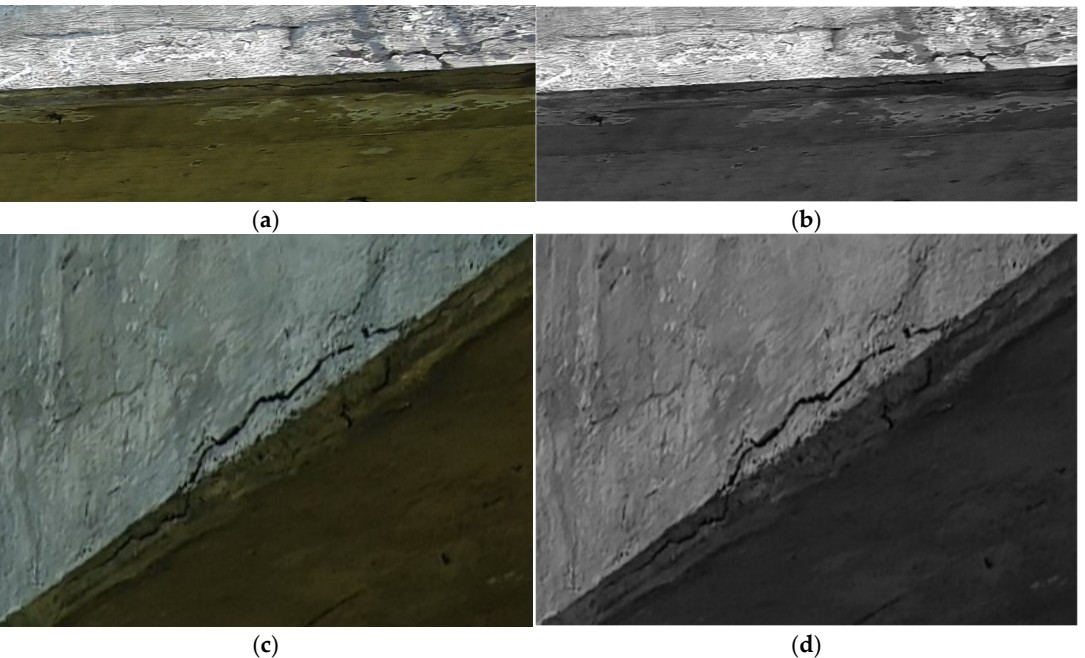

**Figure 6.** Images labeled "MAX_0995" and "MAX_1052" in original RGB format and grayscale. (**a**) Image with label "MAX_0995" RGB. (**b**) Image labeled "MAX_0995" in grayscale. (**c**) Image with "MAX_1052" RGB label. (**d**) Image labeled "MAX_1052" in grayscale.

### 2.2.5. Luminosity Method

This method takes as a reference the perception of brightness of the human eye. Vision is most sensitive to green light, more or less sensitive to red light, and less sensitive to blue light. Therefore, the three variables are accompanied by coefficients that represent their weights according to their wavelengths. Mathematically, it can be expressed through Equation (1), as shown below:

$$Y = 0.299R + 0.587G + 0.114B \tag{1}$$

where Y represents the equivalent value of the grayscale pixel and R, G and B are the values of the red, green and blue color, respectively.

### 2.2.6. Threshold Method Applying the OTSU Technique

The threshold method is one of the most used methods to generate image segmentation. This method allows us to generate binary images that simplify their recognition and classification by reducing the complexity of the image data. This concept is subdivided into two processes: local threshold, and global threshold. On the one hand, local thresholding consists of assigning different threshold values to different regions of the image. On the other hand, the global threshold calculates a general threshold value that will be applied equally to all pixels in the image. As part of our methodology, it is proposed to use the global threshold process [24].

Within the framework of global thresholding, different methods have been proposed, as follows: based on groupings, based on entropy, based on histogram shapes, based on the similarities of object attributes, and based on spatial thresholds. One of the best known and most used in the area of research, that is part of the cluster-based method, is the OTSU technique.

### 2.2.7. OTSU Technique

The OTSU method is used to separate the background from the foreground of the image and is characterized by the following formula:

$$\sigma_B^2 = W_b W_f (U_b - U_f)^2$$

where

$$W_{b,f} = \frac{\text{Number of pixel in the background}}{\text{Total number of pixels}}$$

$$U_{b,f} = \text{mean intensity of the background}$$

The OTSU technique assumes that the image is made up of two kinds of colors: background and object [24]. To achieve this separation between the background and the image of interest, a threshold value, T, must be selected, which will be obtained from the gray-level histogram of the image. Said T value will be the one that classifies the pixels as background or as an object, assigning them a value of 0 and 1, respectively. With f(x,y) being the intensity value of a pixel of a grayscale image and g(x,y) being the value assigned depending on the T given to the pixel, we obtain the following [25]:

$$g(x,y) = \begin{cases} 1, & \text{if } f(x,y) > T \\ 0, & \text{if } f(x,y) \leq T \end{cases}$$

Graphically, it can be visualized and interpreted in Figure 7 [26].

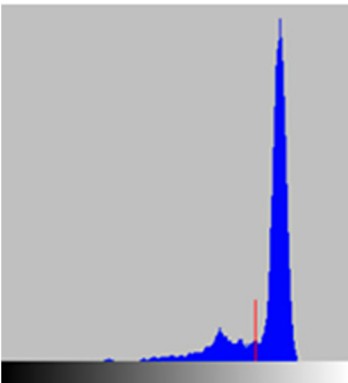

**Figure 7.** Histogram of an image.

In the histogram, the X axis represents the grayscale value and the Y axis represents the number of pixels for each grayscale. Likewise, the red line represents the threshold value, T, which will be responsible for dividing the image pixels between the background and the object.

In Matlab, there is the "graytresh" function that is responsible for providing the threshold value for the image by applying the method mentioned above. Figure 8 shows two histogram results and mean threshold data for two images obtained from the bridge. However, these values are referential. Depending on the code configuration, this threshold value can be set manually to match the gray tones in the image. This was performed to obtain binary images with better highlighted and visible cracks, which can be seen in Figure 9. For example, in the case of the image with label "MAX_0995" seen in Figures 8 and 9, it is reported a large number of pixels corresponding to a grayscale from 60 to 70; that is why a peak is observed in that area of the graph. The red line in the histogram represents the average grayscale of the image based on the number of pixels. That is, the pixels of the image, on average, have approximately a grayscale value of 125. With this average, the algorithm has the reference of granting a threshold value of 0.48 to the binarization process. However, as already anticipated, such a threshold may not be convenient, as it would not highlight the crack. Therefore, a threshold value of 0.1 was assigned manually because, during the image processing, it was observed that the threshold value was too high, causing distortion in the area that needed to be highlighted.

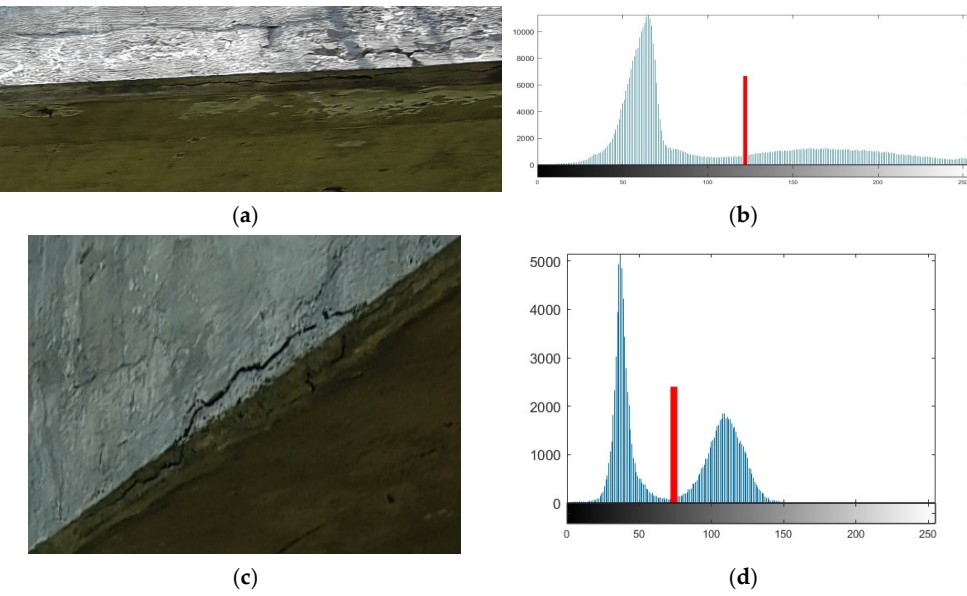

(a)          (b)

(c)          (d)

**Figure 8.** Images labeled "MAX_0995" and "MAX_1052", with their histograms and average threshold. (**a**) Image with label "MAX_0995" RGB. (**b**) Image histogram "MAX_0995" (125 average scale gray and 0.48 threshold). (**c**) Image with "MAX_1052" RGB label. (**d**) Image histogram "MAX_1052" (75 average grayscale and 0.29 threshold).

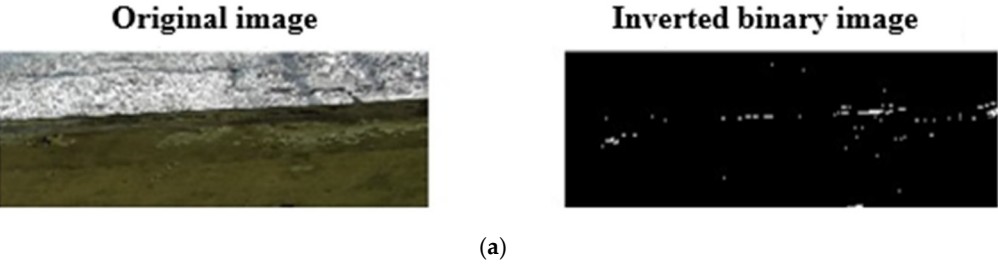

(a)

**Figure 9.** *Cont.*

Original image | Inverted binary image

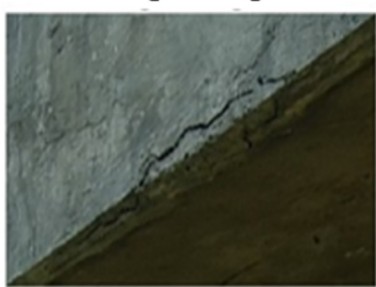 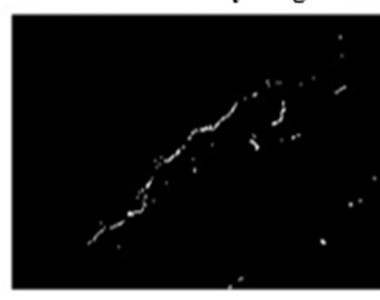

(**b**)

**Figure 9.** Binary segmentation results of images labeled "MAX_0995" and "MAX_1052" and thresholds considered. (**a**) Binary segmentation result of image with label "MAX_0995" with a threshold of 0.1. (**b**) Binary segmentation result of image with label "MAX_1057" with a threshold of 0.08.

Finally, assigning the threshold to the grayscale image is how the binary image is obtained. In Matlab, the "im2bw" function is used for this purpose. The developed algorithms can be adjusted to process folders of images that have similarity in their lighting and color characteristics, with the same threshold value. Thus, it is possible to automate the process.

After binary segmentation, the cracks, being darker than the concrete structures, turn black, while the background turns white. However, for esthetic purposes and ease of understanding, within the Matlab algorithm, it has been contemplated that the blacks and whites can be reversed. In this way, the cracks will appear white on a black background, as seen in Figure 9.

*2.3. Convolutional Neural Networks for Crack Detection*

The CNN, or ConvNet, is a deep learning algorithm that is highly effective for image classification [27]. For the next phase, a CNN is developed, which will be trained with a set of data and, finally, tested to detect damage to the case study bridge.

For training, the following data groups were used: (i) some images of the bridge that did not present cracks (previously selected) that were processed with binary segmentation, (ii) color crack images from other databases that were processed with binary segmentation, and (iii) binary images of cracks from another database. The databases other than the one carried out by this research that corresponds to the Villena Rey Bridge are the following:

- CrackForest Dataset;
- Deteksi Keretakan Pada Struktur Beton by Muhammad Husain;
- "U-Net: Convolutional Networks for Biomedical Image Segmentation" research published in 2015 by Olaf Ronneberger, Philipp Fischer, and Thomas Brox.

For the final testing phase, images exclusively of the bridge that had previously been processed were used. These images had not been used in training, and it was unknown which ones within that group had cracks, since the CNN had to detect them.

The CNN architecture is made up of sequentially stacked layers contained within the Keras linear container. These are as follows: conv2D, Max Pooling, Flatten, Dense, and the output layer. It is important to mention that activation functions have been added for some of the layers, so that the CNN can learn and represent complex non-linear functions. The network works with input images of $100 \times 100 \times 1$—that is to say, binary images with 100 rows and columns. Table 3 details the dimensions of the images with the number of parameters obtained in each layer. Likewise, the matrix of the images, when passing through each of the layers of the architecture, is modified according to the characteristics assigned to each layer. In the final dense layer, thanks to the 'softmax' function, two neurons

are generated that represent the image labels; in other words, it will be responsible for predicting whether the images contain a crack, as an output of the CNN. Figure 10 shows an example of the labeled images.

**Table 3.** Details of the dimensions of the architecture layers.

| Layer | High | Width | Depth | Number of Parameters |
|---|---|---|---|---|
| Input | 100 | 100 | 1 | 320 |
| Conv2D | 98 | 98 | 32 | 320 |
| Max pooling | 49 | 49 | 32 | 0 |
| Flatten | 1 | 1 | 76832 | 0 |
| Dense | 1 | 1 | 15 | 1, 152, 495 |
| Dense_1 | 1 | 1 | 2 | 32 |

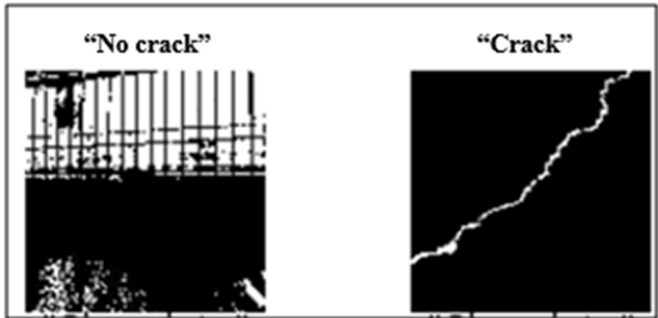

**Figure 10.** Example of the labels assigned to each image.

CNN Considerations for Crack Classification

Firstly, a total of 687 images have been used as input data for training, of which 309 are binary images with cracks and 378 are binary images without cracks. It has been considered to resize the images to a $100 \times 100$ matrix. The sizing of the matrix will depend on the type of images and the architecture developed.

Secondly, for the convolutional layer (Conv2D), 32 filters of size $3 \times 3$ with a stride value of 1 have been used. That is, the convolutional filter moves one pixel at a time over the image pixel matrix. It is worth mentioning that the number of filters may vary depending on the needs of the network. For the max pooling layer, a $2 \times 2$ matrix has been used with a stride value of 2—that is, the pooling matrix moves 2 pixels at a time in both vertical and horizontal directions.

Finally, to configure the CNN processing, a batch size equal to 32, a number of epochs equal to 250, and a learning rate of 0.2 have been considered. It should be noted that the values mentioned are those that are commonly used; however, they can vary depending on the network requirements, and their optimal selection usually requires experimentation and continuous adjustment.

## 3. Results and Discussion

Finally, the CNN is tested, and results are obtained for the structural damage detected in the Villena Rey Bridge, with 88.4% accuracy. Firstly, the application of binary segmentation was evidenced as an effective and viable solution to reduce noise and improve the focus on cracks, thus optimizing the quality of the images for subsequent analysis. Likewise, the CNN, once trained and evaluated through a test phase with exclusive binary images of the bridge that were not part of the training set, is capable of detecting the presence of cracks in these images. As a result, a list of images that identify damage is generated. Since these

images are properly labeled, it is possible to locate these labels within the digital model, thus allowing for the accurate identification of the location of damage in the bridge.

Table 4 shows the list of images with the presence of cracks and their location with respect to the 3D model. As an example, the case of the image labeled "MAX_0963" is taken, which is presented in color and in binary form in Figure 11. There is clearly a crack in this image. To know its location, the Agisoft Metashape Professional should be accessed; then locate in which block of the digital model the photograph is located, activate the "cameras" in the software, select the required image, and the photograph will be highlighted in the corresponding location (Figure 12).

**Table 4.** Record of the images with cracks detected by CNN and their location on the Villena Bridge Rey, taking as reference its location in the digital model.

| Image Tag with Damage Detected | Block of the 3D Model In Which the Image Is Located | Location of Damage on Structure |
|---|---|---|
| MAX_0793 | Block 2 | The image does not correspond to damage |
| MAX_0807 | Block 2 | The image does not correspond to damage |
| MAX_0835 | Block 2 | The image does not correspond to damage |
| MAX_0844 | Block 2 | Left lateral face—Left pillar south area |
| MAX_0845 | Block 2 | Left lateral face—Left pillar south area |
| MAX_0846 | Block 2 | Left lateral face—Left pillar south area |
| MAX_0847 | Block 2 | Left lateral face—Left pillar south area |
| MAX_0848 | Block 2 | Left lateral face—Left pillar south area |
| MAX_0849 | Block 2 | Left lateral face—Left pillar south area |
| MAX_0852 | Block 2 | Left lateral face—Left pillar south area |
| MAX_0912 | Block 2 | Upper face—Left pillar south area |
| MAX_0913 | Block 2 | Upper face—Left pillar south area |
| MAX_0914 | Block 2 | Upper face—Left pillar south area |
| MAX_0960 | Block 2 | Left lateral face—Right pillar south area |
| MAX_0962 | Block 2 | Left lateral face—Right pillar south area |
| MAX_0963 | Block 2 | Left lateral face—Right pillar south area |
| MAX_0964 | Block 2 | Left lateral face—Right pillar south area |
| MAX_1052 | Block 3 | Meeting between right side face and bottom face of the board concrete in the northern area |

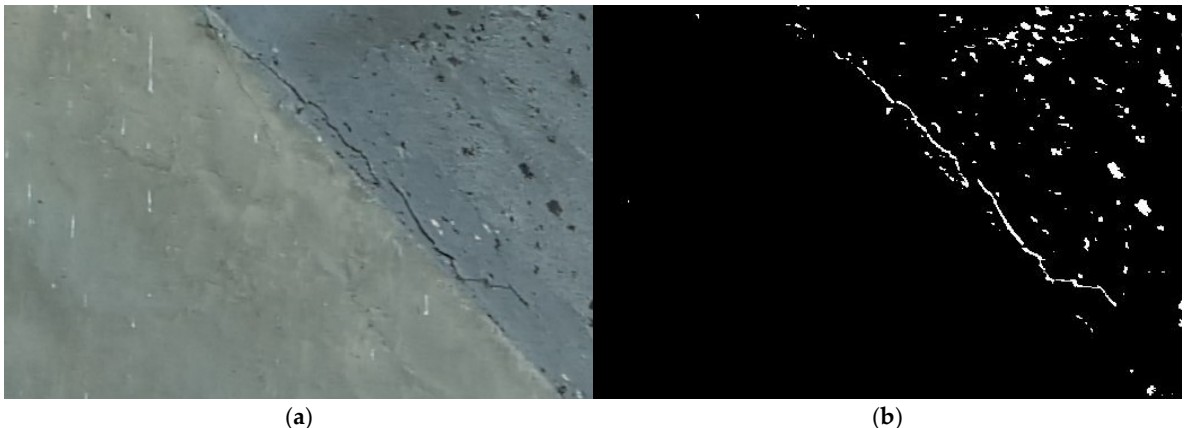

(**a**)                              (**b**)

**Figure 11.** Image "MAX_0912" with a crack detected by CNN. (**a**) Image tagged "MAX_0912". (**b**) Image "MAX_0912" after binary segmentation.

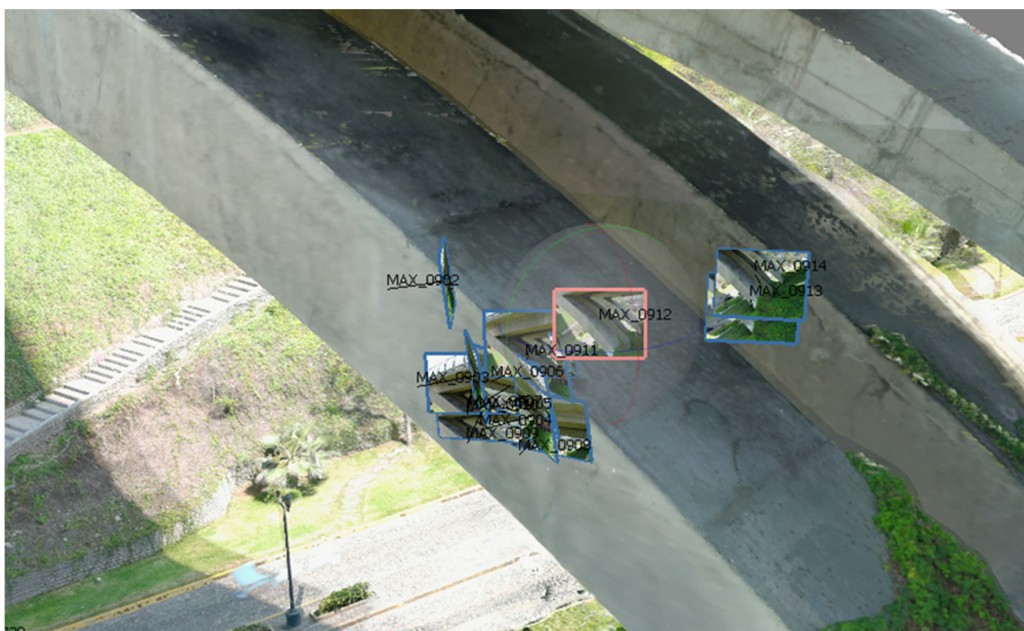

**Figure 12.** Image location "MAX_0912" in the digital model.

However, the accuracy achieved by the CNN was 88.4%. Graphically, the accuracy and losses are shown in Figure 13. On the one hand, it can be observed how the loss decreases as the training iterations progress. Initially, the loss decreases rapidly, indicating that the model is learning and adjusting its parameters effectively. After about 25–30 iterations, the loss stabilizes close to zero, suggesting that the model has learned the patterns in the training data well and reaches a point where more iterations do not provide significant improvements; thus, the previously mentioned range is the ideal number of epochs to train the network. On the other hand, regarding the accuracy vs. iterations graph, accuracy improves rapidly at the beginning, showing that the model is learning to predict correctly. Additionally, very high accuracy values are observed quickly; that is, the model is capable of correctly classifying almost all samples. Finally, the accuracy stabilizes rapidly and remains close to 100%, indicating that the model maintains high performance after the initial iterations.

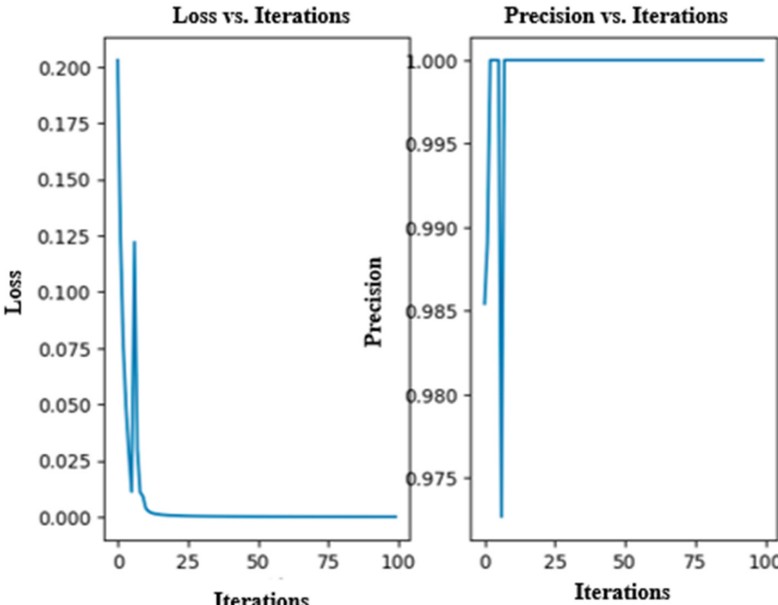

**Figure 13.** Loss vs. graph iterations and precision vs. iterations.

Therefore, it is possible for the CNN to detect a crack in an image that, in reality, does not exist. This is the case for the images "MAX_0793", "MAX_0807", and "MAX_0835", which, as seen in Table 4, do not correspond to damage. This may be due to the fact that these images, in their binary version, can become confusing for the network because of the presence of white continuities that may be interpreted as cracks. What is described can be observed in Figure 14. In the case of Figure 14d, a very clear white line is seen, but if the color version of that image is examined, that continuity is due to the pipe that runs through the bridge structure. Thus, it can also be determined that binary segmentation may emphasize and highlight continuities that are not necessarily cracks. One possible way to correct this detail is by using a database with a larger number of images for the CNN training phase. In this way, there will be more binary images with the presence of white continuities that are not cracks and the CNN, in its training, will detect this feature and will not detect them as cracks. Instead, it will improve the extraction of features from those that are cracks and will identify them more accurately. However, precision is never perfect; therefore, no matter how much training there is, the results must always be corroborated.

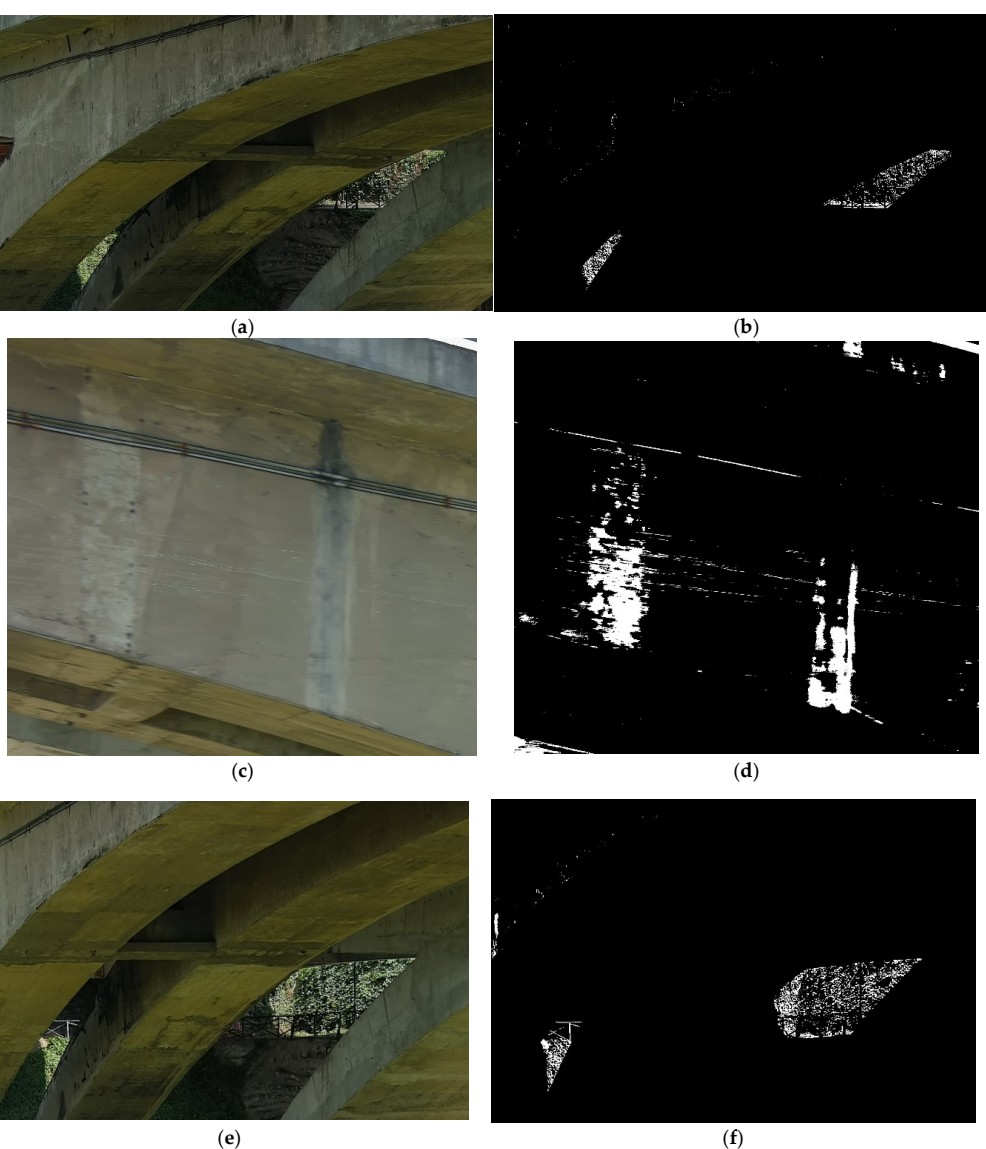

**Figure 14.** Images "MAX_0793", "MAX_0807" and "MAX_0835". (**a**) "MAX_0793" color version. (**b**) "MAX_0793" binary version. (**c**) "MAX_0807" color version. (**d**) "MAX_0807" binary version. (**e**) "MAX_0835" color version. (**f**) "MAX_0835" binary version.

## 4. Conclusions

Currently, visual inspections are the method conventionally used to identify bridge damage; however, this approach has several limitations, such as the long duration of the process and the inherent subjectivity of the evaluation. Research and technological advances have provided civil engineers with new tools and approaches to develop more effective methods for damage detection. These advances facilitate the search for more precise and objective procedures, thus improving the ability to evaluate structural integrity more efficiently.

First, building a three-dimensional model allows us to obtain an accurate digital representation of the real bridge. This model is generated from images captured through a photogrammetric survey using unmanned aerial vehicles (UAV). The images provide a detailed view of the bridge's condition, including areas that are inaccessible to direct human observation. In the project, a digital model composed of five blocks was developed; of these, blocks 2, 3, and 4 were constructed from images close to the concrete elements of the bridge, totaling 397 images in these three blocks. These images were processed using segmentation techniques to improve the visibility of the cracks and reduce noise. Subsequently, the processed images were used as input data in the final testing phase of the convolutional neural network (CNN), responsible for identifying images that have damage.

For this purpose, the images were transformed to grayscale using the luminosity method and binarized using the threshold method and the OTSU technique. The histograms of the images generated a "medium" threshold value for binarization. However, this value served only as a reference, since its application did not always adequately highlight possible cracks. Therefore, assigning the threshold value required an additional manual process to optimize detection.

Third, the deep learning method was used to develop convolutional neural networks for crack detection in binary images. On the one hand, the data used for training the CNN was a total of 687 images outside the study bridge of $100 \times 100$ pixel resolution between binary images with cracks and without cracks. On the other hand, in the prediction stage, the input data for the network were 129 binary images corresponding to the case study. An accuracy of 88.4% was achieved and 18 images with the presence of damage were detected. However, due to the precision, it was necessary to view these images to really verify that they have damage. In this way, it was determined that the images "MAX_0793", "MAX_0807", and "MAX_0835" did not have cracks, but their binary versions did have white continuities that could confuse the network.

In summary, an agile and, above all, precise method is achieved to detect visual damage in reinforced concrete bridges, through the construction of a 3D model and the use of deep learning. The accuracy of 88.4% is acceptable, since methods based on damage detection with images obtained with UAVs must be around 90% [11]. The good results of the method are justified by the correct training of the CNN, where binary images from other databases were also used to increase the training capacity, and in the binary segmentation process that allows for the CNN to have greater fluidity when detecting cracks. However, the accuracy can be improved with training with a larger amount of data and achieve even more acceptable and favorable results compared to other research that manages to exceed 95% accuracy [12,21]. Finally, damage inspection engineers can use this method in a simple and practical way by having a UAV and cameras with minimum and adequate characteristics, such as those specified in this research, and by correctly using binary segmentation and CNN algorithms. As a recommendation, the 3D model should be updated by building a new one on an annual basis, since the Peruvian Bridge Inspection Guide states that bridges in service should be assessed at least once a year. However, it is also valid to carry out more expeditions with UAVs to update the 3D model more than

once a year and thus monitor its damages in a more present manner. It is worth mentioning that the method is intended to capture a "current state" of a bridge. While it is true that 3D models can be obtained and damage detection applied with the developed CNN at different times, the proposed 3D model is not intended to be a "live" model, but rather a "specific" one.

**Author Contributions:** Conceptualization, M.C.A. and R.S.V.; methodology, M.C.A., R.S.V. and R.M.D.; software, M.C.A. and R.S.V.; validation, J.R.C., R.M.D. and L.M.; formal analysis, M.C.A., R.S.V. and R.M.D.; investigation, R.M.D., M.C.A. and R.S.V.; resources, R.M.D., M.C.A. and R.S.V.; data curation, M.C.A., R.S.V. and R.M.D.; writing—original draft preparation, M.C.A., R.S.V. and R.M.D.; writing—review and editing, J.R.C., R.M.D. and L.M.; visualization, J.R.C. and R.M.D.; supervision, J.R.C., R.M.D. and L.M.; project administration, R.M.D. All authors have read and agreed to the published version of the manuscript.

**Funding:** This research was funded by Consejo Nacional de Ciencia, Tecnología e Innovación Tecnológica (CONCYTEC) y el Programa Nacional de Investigación Científica y Estudios Avanzados (PROCIENCIA) en el marco del concurso "E067- 2023-01 Proyectos Especiales: Proyectos de Incorporación de Investigadores Postdoctorales en Instituciones Peruanas" grant number PE501084691-2023.

**Data Availability Statement:** Data not published due to privacy restrictions.

**Acknowledgments:** The authors would like to express their gratitude and funding provided by the Consejo Nacional de Ciencia, Tecnología e Innovación Tecnológica (CONCYTEC) y el Programa Nacional de Investigación Científica y Estudios Avanzados (PROCIENCIA) en el marco del concurso "E067- 2023-01 Proyectos Especiales: Proyectos de Incorporación de Investigadores Postdoctorales en Instituciones Peruanas" grant number PE501084691-2023.

**Conflicts of Interest:** The authors declare no conflict of interest.

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
