# Peer review of "Structural Damage Detection Using an Unmanned Aerial Vehicle-Based 3D Model and Deep Learning on a Reinforced Concrete Arch Bridge"

_infrastructures, doi:10.3390/infrastructures10020033_

Round 1
Reviewer 1 Report
Comments and Suggestions for Authors
Please see the attached file for comments.

Author Response
- In line 24 of page 1, it is said that “…, improving the accuracy and efficiency of damage identification”. There is no comparison of methods in the manuscript. How did the author reach this conclusion?
The sentence is based on the comparison with traditional damage detection methods as the visual inspection. In order to clarify the meaning we have added this explanation in the abstract
In the conclusions, it is mentioned that the accuracy percentage is 88.4%, without comparing with the accuracy of other damage detection methods.. This value is acceptable and close to the 90% accuracy found in other articles that analyze images with UAVs, as mentioned in the article “Measurement of Cracks in Concrete Bridges by Using Unmanned Aerial Vehicles and Image Registration.” Furthermore, it is added that accuracy can be improved by using a larger amount of data, potentially exceeding approximately 95%. This is referenced in the articles “Measurement of Cracks in Concrete Bridges by Using Unmanned Aerial Vehicles and Image Registration” and “Automatic recognition system for concrete cracks with support vector machine based on crack features,” which are also cited within the paper. (A comment is added in the updated paper.)
- In line 75 of page 2, the citation format of the manuscript needs to be standardized, it is recommended to modify it to “Perez Jimeno et al. [9] carried out UAV flights to build a 3D model of the Río Claro Bridge (Colombia)”. A similar problem exists on lines 77 to 111.
We have fixed the citation format according to the suggestion of the reviewer in the whole paper.
- In Figure 3 in line 204 of page 6, in the Binary Segmentation part, the flowchart of the proposed method needs to be supplemented
A prior explanation is provided in the paragraph on line 181 (a comment is added in the updated paper).
- In line 249 of page 7, in this part, a detailed description of the UAV flight paths is required.
. Added in the revised version of the paper. (A comment is added in the updated paper.)
- In Figure 5 d) in line 279 of page 8, the bottom of the main beams is very limited when viewed from this perspective. For example, the cracks in Figure 2 are not visible from this perspective. Why doesn’t the flight path pass under the bridge?
In Figure 5.d, two arches can be observed, each corresponding to a bridge: the first arch (which is shown as limited) corresponds to the Mellizo Bridge (the new bridge), and the second arch (which is better constructed in the 3D model) corresponds to the Villena Rey Bridge (old), that is, the bridge under study. The difference between both bridges is explained in section “1.1. Case Study.”
On the other hand, if you zoom in on the lower part of the bridge in block 4, as referenced in Figure 5.d, some details of its composition can be observed. However, it is true that there are certain parts of the bridge where the UAV has not been able to access comfortably 100%, one of those areas being underneath the bridge, as the UAV may lose its geolocation.
Figure 2 is more of a way to justify that the old Villena Rey Bridge has damages and is therefore a good case study for applying the method.
- In line 332 of page 10, the sequence number of title is repeated. A similar problema exists on line 362.
.OK. This is corrected in the new version of the paper. (A comment is added in the updated paper.)
- In line 396 of page 12, the reason for setting the threshold to 0.48 needs to be explained, and the formula for calculating the threshold needs to be given
The OTSU method is used to separate the background from the foreground of the image and is characterized by the following formula:
where
Therefore, by applying the OTSU method algorithm in Matlab, the parameter ​ will provide the threshold value for the pixels that will be classified as background and foreground, which needs to be highlighted. Thus, in Matlab, a value of 0.48 was obtained. This is not explained in the revised paper
- In line 398 of page 12, it is necessary to explain the rationality of manually updating the threshold to 0.1. What is the basis? How to avoid misjudgment caused by reducing the threshold?
As mentioned earlier, the OTSU method provides an average value that separates the background from the foreground by assigning them values of 1 and 0. However, during the image processing, it was observed that the threshold value was too high, causing distortion in the area that needed to be highlighted. Therefore, it was necessary to manually adjust this value to suit the conditions of the images since the lighting and angles at which they were captured did not adequately emphasize the cracks. Additionally, to determine the final threshold value, the value obtained from Matlab was used as a reference to guide how much the value could be adjusted.
- In Figure 8 b) and d) in line 410 of page 12, the units of the coordinate axes need to be supplemented.
In line 378 of page 11, the text describes what each axis of the histogram represents: “…the X axis represents the grayscale value and the Y axis represents the number of pixels for each grayscale.” (a comment is added in the updated paper).
- In Figure 9 a) in line 412 of page 13, the cracks in the original image are not obvious in the inverted binary image. The reason needs to be explained
In the original images, preprocessing was carried out to better visualize the crack within the image and to achieve the inverted binary image. As part of this process, efforts were made to enhance the crack in a way that it appears uniform by assigning values that are suitable for the characteristics of the photographs.
- In line 416 of page 13, what improvements did the author make to the CNN model? Compared with other CNN models, how is the accurate and efficient of the proposed method?
As observed in line 425, the inverted binary images have been used as input data for the CNN, allowing for more efficient processing in damage detection. Therefore, these enhancements are considered part of the improvements made during the development of the neural network. Additionally, as mentioned in comment 1, various authors have demonstrated that the accuracy percentage regarding the use of a CNN for crack detection with UAVs is high and acceptable. (A comment has been added in the updated paper.)
- In Figure 13 in line 516 of page 17, the reason for the sudden change in the figure needs to be explained.
In line 492 of page 15, there is a mention of the two graphs (Figure 13) obtained from the image processing with the CNN. Therefore, both images represent what is explicitly stated, complementing each other. (A comment has been added in the updated paper.)
- In line 537 of page 19, what is the minimum crack width that the proposed method can identify? In addition, cracks will develop. How often should this 3D model be updated? When updating the 3d model, is a new 3d model created? Or is only the part with crack updated?
(i) The minimum crack width that the proposed method can identify is 0.4 mm. According to Sotomayor, referenced in the paper as “[17],” the minimum width for a crack to have structural implications is also 0.4 mm, and our method aims to detect damages, not common fissures. The reason for asserting that this is the minimum identifiable width is that, due to the camera's distance, a UAV would not be able to capture fissures narrower than this. Therefore, the method only captures cracks that have structural significance.
(ii) The Peru Bridge Inspection Guide states, “Bridges in service must be evaluated at least once a year by personnel specifically trained for the identification and evaluation of damages” (p. 8). Therefore, if visual inspections for damage detection are currently performed annually, it is also recommended to update the 3D model annually. However, since the proposed method is more efficient and ensures accuracy, it is suggested to update it even more frequently.
(iii) Since this involves photogrammetry, it is recommended to update the 3D model with a completely new one. For the model to integrate correctly, all photographs must be interconnected accurately through geolocation. When integrating new photographs not originally considered, the interrelation of data may vary. It is important to mention that the proposed method is designed for damage detection at a specific moment, meaning for periodic inspections, rather than for constant updates. The 3D model aims to project and represent a “current state”; it is not intended to be a “live” representation.

Reviewer 2 Report
Comments and Suggestions for Authors
The study has built a three-dimensional model that allows one to obtain an accurate digital representation of a real bridge. This model is generated from images captured through a photogrammetric survey using unmanned aerial vehicles (UAV). This study also proposed an effective approach to identify cracks of the real bridge using photographs obtained by an UAV and the use of a convolutional neural network (CNN). Although the identification results are acceptable, there are still areas that need to be explained or improved.
1. The photos in Figure 2 should be listed in the correct orientation and should indicate which part of the bridge the photos correspond to.
2. The text should clearly indicate what the input and output of CNN are.
3. The authors state that "some binary images of the bridge and additional binary images from other databases will be used for training the CNN.". The database information of other color and binary images for training CNN should be given.
4. The study produced a 3D model of the bridge from a limited collection of photos taken by the UAV. How does the photo correspond to the part or location of the bridge being photographed? How to splice photos in the same block effectively?
5. The original photo in Figure 9 doesn't seem very clear. When taking a photo, what is the distance between the UAV and the bridge part? How do you guarantee the quality of all the photos?
6. Why do you select 0.1 as threshold value for the binarization? Can it highlight the crack? Why?
7. Figure 13 is of poor quality. Is it a loss curve and accuracy curve for CNN training? Please state precisely.
Author Response
Reviewer 2
The study has built a three-dimensional model that allows one to obtain an accurate digital representation of a real bridge. This model is generated from images captured through a photogrammetric survey using unmanned aerial vehicles (UAV). This study also proposed an effective approach to identify cracks of the real bridge using photographs obtained by an UAV and the use of a convolutional neural network (CNN). Although the identification results are acceptable, there are still areas that need to be explained or improved.
- The photos in Figure 2 should be listed in the correct orientation and should indicate
Corrected (a comment is added in the updated paper).
- The text should clearly indicate what the input and output of CNN are.
The CNN consists of two phases: the training phase and the final testing phase.
On one hand, as mentioned in line 456 of page 15: “Firstly, a total of 687 images have been used as input data for training, of which 309 are binary images with cracks and 378 are binary images without cracks. It has been considered to resize the images to a 100 × 100 matrix. The sizing of the matrix will depend on the type of images and the architecture developed.” This means that binary images with and without cracks were used as input data for training the CNN. It is important to highlight that the training has no "outputs"; it focuses solely on allowing the CNN to distinguish between a binary image with a crack and a binary image without a crack.
This idea is reinforced in line 427 of page 13, where it states, “For training, the following data groups were used: (i) some images of the bridge that did not present cracks (previously selected) that were processed with binary segmentation, (ii) color crack images from other databases that were processed with binary segmentation, and (iii) binary images of cracks from another database.” This specifies the sets of images that served as input data for the CNN during its training phase.
On the other hand, in line 430 of page 13, it is indicated: “For the final testing phase, images exclusively of the bridge that had previously been processed were used. These images had not been used in training, and it was unknown which ones within that group had cracks, since the CNN had to detect them.” Therefore, for the final testing phase, the input data consisted of processed binary segmentation images of the Villena Bridge that had not been used in the training phase.
Similarly, in line 442 of page 13, it states, “…it will be responsible for predicting whether the images contain a crack.” This means that the output data of the convolutional neural network will be a summary chart showing the images that contain cracks and those that do not. Additionally, a reinforcing idea is included. (A comment has been added in the updated paper.)
- The authors state that "some binary images of the bridge and additional binary images from other databases will be used for training the CNN.". The database information of other color and binary images for training CNN should be given.
For training the convolutional neural network, we used photographs of cracks extracted from various databases:
- CrackForest Dataset: This is a database that provides photographs of cracks in pavements.
- Deteksi Keretakan Pada Struktur Beton by Muhammad Husain: This article served as a source for its database to complement the training data for the neural network.
- From the paper “U-Net: Convolutional Networks for Biomedical Image Segmentation,” published in 2015 by Olaf Ronneberger, Philipp Fischer, and Thomas Brox, binary images of concrete surfaces were extracted for use in the training process of the CNN.
- The study produced a 3D model of the bridge from a limited collection of photos taken by the UAV. How does the photo correspond to the part or location of the bridge being photographed? How to splice photos in the same block effectively?
The UAV, when taking a photograph, records its exact location, which is necessary for performing a process known as "photogrammetry." This location is encoded within the 3D model in the Agisoft software, as visually depicted in Figure 12. This way, it is possible to identify which element of the bridge each photograph refers to. This idea has been added in the updated paper. (A comment has been added in the updated paper.)
Regarding the observation questioning "how can the photographs be combined into a single block," as detailed in the article, the generated 3D model consists of 5 blocks or 5 submodels. Since each image is tied to a specific location, the photogrammetric processing to integrate all images can sometimes be resource-intensive, prompting the decision to create "blocks." For this research, it is not necessary to merge all photographs into a single block; although the blocks are “independent,” they are referenced for the same case study (the Villena Bridge). It is only important to identify which elements of the bridge each block refers to.
However, for future research or applications of the 3D model, it is worth noting that it is possible to integrate all photographs into a single block through the photogrammetric processing of the software, even though this may be resource-intensive.
- The original photo in Figure 9 doesn't seem very clear. When taking a photo, what is the distance between the UAV and the bridge part? How do you guarantee the quality of all the photos?
The approximate distances for taking photographs of the structural elements of the bridge are between 2 to 3 meters. It is advisable not to bring the UAV too close to the bridge, as it may destabilize and collide due to air turbulence, despite meeting the minimum standards outlined in line 233 on page 7.
The quality of the photographs is ensured by adhering to the minimum camera standards specified in line 237 on page 7. (A comment has been added in the updated paper.)
- Why do you select 0.1 as threshold value for the binarization? Can it highlight the crack? Why?
As indicated in line 396 on page 12, the threshold value CAN be obtained from the grayscale histogram corresponding to the OTSU technique. However, it is also mentioned that this value can be manually "adjusted" to enhance the visibility of cracks or further reduce noise in the resulting binary image. Therefore, an example is provided with the images in Figures 8 and 9, where threshold values are adjusted until better results are achieved. (A comment has been added in the updated paper.)
To respond to the query, the value of 0.1 was selected for the example in Figure 9.a. because it yielded the best results. This was derived from the value of 0.48 obtained from the histogram (Figure 8.a.) of grayscale, and the threshold was subsequently adjusted from that point.
- Figure 13 is of poor quality. Is it a loss curve and accuracy curve for CNN training? Please state precisely.
The graphs in Figure 13 represent the results of the CNN, showing both iterations versus accuracy and loss during the crack detection phase of the bridge images. This information is detailed in line 494 on page 15. (A comment has been added in the updated paper.)
